# Insights into the Genetic Diversity of *Leishmania (Viannia) panamensis* in Panama, Inferred via Multilocus Sequence Typing (MLST)

**DOI:** 10.3390/pathogens12050747

**Published:** 2023-05-22

**Authors:** Daniel Mendieta, Vanessa Vásquez, Luis Jaén, Vanessa Pineda, Azael Saldaña, José Eduardo Calzada, Franklyn Samudio

**Affiliations:** 1Facultad de Ciencias Naturales Exactas y Tecnología, Universidad de Panamá, Panama City P.O. Box 0824-00073, Panama; daniel.mendieta@up.ac.pa (D.M.); luisangeljaen@gmail.com (L.J.); azasal@hotmail.com (A.S.); 2Instituto Conmemorativo Gorgas de Estudios de la Salud, Avenida Justo Arosemena, Panama City P.O. Box 0816-02593, Panama; vvasquez@gorgas.gob.pa (V.V.); vpineda@gorgas.gob.pa (V.P.); jcalzada@gorgas.gob.pa (J.E.C.); 3Centro de Investigación y Diagnóstico de Enfermedades Parasitarias (CIDEP), Universidad de Panamá, Panama City P.O. Box 0824-00073, Panama; 4Sistema Nacional de Investigación, Panama City P.O. Box 0816-02852, Panama

**Keywords:** tegumentary leishmaniasis, *Leishmania panamensis*, multilocus typing, diploid sequence type, Panama

## Abstract

Leishmaniasis is a disease caused by parasites of the genus *Leishmania* and transmitted by sand fly vectors. Tegumentary leishmaniasis is the most prevalent clinical outcome in Latin America, afflicting people from 18 countries. In Panama, the annual incidence rate of leishmaniasis is as high as 3000 cases, representing a major public health problem. In endemic regions, *L. panamensis* is responsible for almost eighty percent of human cases that present different clinical outcomes. These differences in disease outcomes could be the result of the local interplay between *L. panamensis* variants and human hosts with different genetic backgrounds. The genetic diversity of *L. panamensis* in Panama has only been partially explored, and the variability reported for this species is based on few studies restricted to small populations and/or with poor resolutive markers at low taxonomic levels. Accordingly, in this study, we explored the genetic diversity of sixty-nine *L. panamensis* isolates from different endemic regions of Panama, using an MLST approach based on four housekeeping genes (Aconitase, *ALAT*, *GPI* and *HSP70*). Two to seven haplotypes per locus were identified, and regional differences in the genetic diversity of *L. panamensis* were observed. A genotype analysis evidenced the circulation of thirteen *L. panamensis* genotypes, a fact that might have important implications for the local control of the disease.

## 1. Introduction

Leishmaniasis comprises a group of diseases caused by protozoan parasites of the genus *Leishmania* and transmitted by sand fly vectors with an eminent neglected status. The main clinical forms of the disease are visceral (VL) and tegumentary leishmaniasis (TL), which manifest different clinical expressions depending on the causative species and its genetic background in combination with the immunological status of the host and factors in the sand fly’s saliva [1,2]. This disease has been considered an emerging/remerging disease by the World Health Organization (WHO) because of the high and increasing number of cases and its geographical expansion [3]. Leishmaniasis is endemic in 98 countries worldwide, having an estimated annual incidence of 2.0 million cases and an approximate prevalence of 12,000,000 cases [4]. Additionally, this disease causes around 20,000 to 40,000 deaths in endemic areas all year round [3,4]. 

In the Americas, TL is endemic in 18 countries in which this clinical form is 15 times more frequent than VL [3]. In Panama, leishmaniasis is the most prevalent vector-borne parasitic disease, reaching an incidence rate as high as 3000 annual cases and afflicting people of all ages. The local transmission of leishmaniasis occurs mainly in rural and sylvatic environments, affecting economically marginalized people with TL [3]. 

At least twenty species are included in the *Leishmania* genus, which is divided into the sections Euleishmania and Paraleishmania, which are constantly under taxonomical review [5]. In Panama, *L. panamensis* is the principal etiological agent of TL [6,7]. However, other *Viannia* species, such as *L. braziliensis*, *L. guyanensis,* and *L. naiffi*, have been found circulating sympatrically at minor frequencies, also causing TL [8]. Additionally, a few cases of TL caused by *L. mexicana* have been reported in the country [6,9]. The initial reports of *Leishmania* species in the country were made using the gold standard approach, multilocus enzyme electrophoresis (MLEE), and after the nineties, reports were made via molecular protocols based on PCR-RFLP or PCR sequencing of specific genes such as calmodulin [10,11], *HSP70* [8], cytochrome b [12], and kinetoplast minicircles [7]. Among the *Leishmania* species reported using these tools, *L. panamensis* was the most prevalent species found in human cases of TL in Panama. In Latin American countries, *L. panamensis* is responsible for multiple clinical forms such as localized, disseminated, mucocutaneous, and diffuse forms of the tegumentary disease [13,14,15,16]. These clinical outcomes could be related to the local interplay between genetic variants of this species and the complex local population comprising human inhabitants with different genetic backgrounds. Thus, the plethora of clinical forms caused by this species and other medically relevant characteristics within the *L. panamensis* population need to be addressed. To perform this task, it is initially important to assess the local genetic variability of this species using more informative molecular tools capable of finding intra-specific variations. In the case of the *Leishmania* genus, single-locus analyses are sometimes not sufficient to find intra-specific variations and methodologies that add more loci to the analysis should be chosen instead to increase the discriminatory power. Studies based on multilocus sequence typing (MLST) in *Leishmania* have demonstrated that this technique has more resolutive power than the gold standard, MLEE, to study intra-specific genetic diversity in *Leishmania* [17,18]. Furthermore, a great level of diversity has been shown when a high number of isolates from *Leishmania* species have been analyzed using MLST, supporting the use of MLST as a tool to study intra-specific variability within *Leishmania* parasites [19,20,21]. Regarding multilocus approaches, only one molecular tool based on multilocus microsatellite typing (MLMT) has been previously used to evaluate a small number of *L. panamensis* isolates from the Central region of Panama, revealing the apparent local genetic diversity of this *Leishmania* species [22]. However, to elucidate the real genetic diversity of this species in Panama, it is necessary to evaluate isolates from the main endemic areas of the country, applying tools with high resolutive power at lower taxonomic levels. Considering this, in this study, we evaluated sixty-nine *L. panamensis* isolates from the main endemic areas of leishmaniasis in the country using an MLST approach based on four essential genes. Our findings suggest the circulation of different *L. panamensis* variants in the country and evidence of regional differences in genetic diversity, a fact that could be related to the different clinical outcomes caused by this species. The knowledge gathered herein might also provide a baseline for developing further studies on medically relevant characteristics that impact the clinical outcome of the disease as well as resistance to the first-line drugs used to treat leishmaniasis.

## 2. Materials and Methods

### 2.1. Leishmania Isolates

Sixty-nine *Leishmania panamensis* isolates from both the eastern border and western border of the Panama Canal were used in this study. The *Leishmania* isolates were collected over the last five years (2015–2020) and stored in liquid nitrogen at the Dirección de Investigación en Parasitología, Instituto Conmemorativo Gorgas de Estudios de la Salud (ICGES), situated in Panama City. The parasites were isolated from patients living in provinces located east (Panama and Darien) and west (Colon, Panama Oeste, Cocle, and Bocas del Toro) of the Panama Canal (Figure 1). These locations are recognized as endemic areas of leishmaniasis in the country, with a long-term average CL incidence rate of between 5 and 25 new cases per 10,000 inhabitants [23].

### 2.2. Parasite Culture and DNA Extraction

*Leishmania* stocks were cultured using Schneider’s insect medium (Sigma Aldrich, Inc., St. Louis, MO, USA) supplemented with 20% heat-inactivated fetal bovine serum (Gibco, Grand Island, VT, USA) at 26 °C. Total genomic DNA was extracted from the promastigotes using a commercial kit (Wizard Genomic DNA Purification Kit, Promega, Madison, WI, USA) according to the manufacturer’s instructions.

### 2.3. Molecular Identification of Leishmania panamensis Isolates

#### 2.3.1. PCR Amplification

DNA samples from parasite cultures were used to identify *L. panamensis* parasites through an approach based on the PCR amplification and sequencing of a 1364 bp segment from the *Leishmania HSP70* gene. To perform this PCR, we designed the set of primers, PLeishF 5′-GATGGTGCTGCTGAAGATGA-3′ and PLeishR 5′-GGTCATGATCGGGTTGCATR-3′, that amplify the *HSP70* gene from the main *Leishmania* complexes, using the primer3 algorithm included in the UGENE bioinformatic platform. The specificity of the primers was evaluated using PrimerBlast software (https://www.ncbi.nlm.nih.gov/tools/primer-blast/, accessed on 1 January 2021), and the formation of inter- and intra-molecular secondary structures was assessed using the OligoAnalyzer tool (https://www.idtdna.com/calc/analyzer, accessed on 1 February 2021). The DNA samples were amplified in a final volume of 50 µL containing 10 µL of 10× buffer, 200 µM of DNTPs, one unit of Phusion DNA polymerase, 0.3 µM of each primer, 50 nanograms of genomic DNA, and nuclease-free water until a total volume of 50 µL was reached. The thermal conditions of the PCR reactions were an initial denaturation at 98 °C for 30 s, followed by 34 cycles at 98 °C for 10 s, 65 °C for 30 s, 72 °C for 30 s, and a final extension step of 72 °C for 5 min.

#### 2.3.2. Sanger Sequencing and Species Discrimination via Phylogenetic Analysis

The purified products were sequenced in both senses using the BigDye™ Terminator v3.1 Cycle Sequencing Kit and analyzed in the ABI 3500XL genetic analyzer (Applied Biosystems, CA, USA). The electropherograms of the *HSP70* gene fragment were then assembled in both senses using the assembled-to-reference tool of the UGENE software v.39 to obtain HSP70 consensus sequences in the FASTA format. All the *HSP70* sequences obtained in this study were submitted to the GenBank database, where they can be retrieved using the accession numbers OQ408828 to OQ408896. Afterward, the consensus sequences were aligned using the MAFFT algorithm included in the UGENE tool, and this alignment was used to find the best substitution model via JModelTest [24]. The GTR + G model then was used in Mr. Bayes v.6.12.03 software to perform a phylogenetic construction that helped allocate isolates within specific *Leishmania* species clusters. To accomplish this task, twenty Markov chains were used for six million generations, and trees were sampled every ten thousand generations. The HSP70 sequence CP015689.1, extracted from chromosome 39 of the *Trypanosoma cruzi* Sylvio strain, was used as an outgroup to construct the phylogenetic tree. Twenty-five percent of the sampled trees were discarded, and the remaining were used to build a consensus tree and calculate the clades’ posterior probabilities. The results of the Bayesian analyses were visualized using Figtree v1.4.2 (FigTree (ed.ac.uk)). The transformation of the leaves and a schematic representation of the root were applied for visualization purposes.

### 2.4. PCR Amplification and Sequencing of Gene Loci Used in the MLST Approach

To infer the genetic diversity of *L. panamensis* isolates, we used an MLST approach based upon the amplification and sequencing of specific gene fragments from the metabolic enzymes *GPI*, aconitase, and *ALAT* using the gene-specific primers depicted in Table 1. In this table also the gene length, chromosomal location, and product size of each gene locus used in this study are also depicted. Additionally, the *HSP70* sequences obtained via the *Leishmania* species discrimination approach were used in the MLST scheme. These four markers appear orthologous throughout the *Leishmania* genus, and three of them codify metabolic enzymes generally used to type *Leishmania* via multilocus enzyme electrophoresis (MLEE). Additionally, the aconitase, *ALAT*, and *GPI* loci have shown several polymorphisms that aided in the exploration of the intraspecific and/or interspecific genetic variability of *Leishmania* species at the haplotype and genotype levels when using reference sequences and/or sequences of *Leishmania* isolates from some Latin American countries [21,25]. Regarding the HSP70 gene, it has been added successfully to MLST schemes to study the genetic diversity of *L. braziliensis* and *L. panamensis* in Colombia [26] and to study *L. braziliensis* outbreaks in Brazil [25]. In these studies, all four markers identified several haplotypes and showed a high level of discriminatory power in determining the diploid sequence types (DSTs) of the *Leishmania viannia* species. Considering the facts presented above, we decided to explore the genetic diversity of *L. panamensis* in Panama using the abovementioned loci.

All PCR reactions were performed in a final volume of 50 µL, including 5 µL of 10× buffer, 2.5 µL of MgSO4, 200 µM of dNTPs, one unit of Taq polymerase, and 50 ng of genomic DNA. The primer concentrations for each set were 0.4 µM for the *ALAT* and aconitase primers and 0.3 µM for the *GPI* set of primers. The target DNA was amplified after a primary denaturation at 95 °C for 5 min, followed by 35 cycles consisting of 95 °C for 30 s, an aligning temperature of 55 °C for aconitase/*ALAT* genes and 53 °C for the *GPI* fragment, and a target extension of 68 °C for 90 s. The PCR products were then extended at 68 °C for 10 min as a final PCR step. Five microliters of all amplicons were analyzed in 1.2% agarose gel stained with GelRed for 45 min at 85 V. The rest of the amplicons were purified using the commercial kit Wizard SV gel and the PCR Clean-Up System, following the manufacturer’s instructions (Promega, Madison, WI, USA). The amplification products then were sequenced in both senses using the BigDye™ Terminator v3.1 Cycle Sequencing Kit and analyzed in the ABI 3500XL genetic analyzer (Applied Biosystems, CA, USA).

### 2.5. Sequence Assembling, Haplotype Construction, and Determination of Diversity Indexes

We used the map reads of the reference tool included in the Sanger data analysis option of the bioinformatic software UGENE v.39 [28] to edit and assemble the electropherograms, using both senses in order to obtain consensus sequences from each molecular marker. The consensus sequences obtained from the aconitase, *ALAT*, and *GPI* loci were placed into the GenBank online servers under the range of accession numbers from OQ408621 to OQ408827. The DNAsp software v.6.12.03 [29] was subsequently utilized to construct *L. panamensis* haplotypes via the PHASE algorithm, setting the MCMC options of the coalescent-based Bayesian method to use 1000 interactions, 1 thinning interval, and 100 burn-in interactions. After constructing the haplotypes, we used their sequences to infer the genetic diversity indexes, including the total number of mutations (ETA), haplotype diversity (Hd), nucleotide diversity (π), and diversity index ϴ, using the same bioinformatic software. To determine the diversity indexes, the number of segregation sites, and haplotype number by the geographical area evaluated, we independently assessed sequences of all four loci from 34 and 35 *L. panamensis* isolates belonging to the western and eastern regions of the country through the use of DNAsp v6.12.03. 

### 2.6. Determination of L. panamensis DSTs and Their Geographical Distribution

The bioinformatics software MLSTest [30] was used to (1) determine the best number and combinations of loci that guaranteed to find the highest diversity indexes of the local isolates of *L. panamensis*, (2) analyze incongruence among the loci chosen for the MLST approach, and (3) calculate the efficiency and discriminatory power of the MLST scheme. To select the number of loci that offered the highest diversity index, we used the scheme optimization options of the software to reveal whether the genotypic diversity increased as more loci were added to the MLST scheme. Regarding the congruency analyses, we searched for incongruency by determining the *p*-value for the incongruence length difference parameters (ILDs) of all selected loci. Furthermore, we applied the Templeton test to evaluate incongruency between all loci and the topology of the concatenated tree, setting the algorithm to make an overall analysis of incongruency. To obtain the allelic profiles and the different types of *L. panamensis* DSTs, we used the average state option of the MLSTest program to handle the heterozygous sites and calculate the genetic distances. in addition, we obtained the typing efficiency and the discriminatory power for each locus after using the option to make/view the allelic profile in the same program. 

Using the software QGIZ 3.16 Hannover (https://qgis.org, accessed on 1 March 2022), a map of the local distribution of *L. panamensis* DSTs was constructed based on the geographical coordinates of each place at which the DSTs were found.

### 2.7. Phylogenetic Analyzes of L. panamensis DSTs

To corroborate the *L. panamensis* genotypes by phylogenetic analysis, we selected representative sequences from each DST found herein. We phased the sequences from each representative DST via DNAsp software v.6.12.03, and the alleles obtained were used to perform a Bayesian-based phylogenetic analysis. Allelic sequences from each locus used were concatenated using the software SeaView version 1:4.5.4.8-2 [31]. Sequences were then multiply aligned using the MAFFT algorithm, which is also included in UGENE, with a maximum number of iterative refinements of three and a gap penalty of 1.53. The HKY (G + I) model was identified as the best DNA evolution model by the software JModelTest [24]. A phylogenetic tree reconstruction of *Leishmania* was implemented by applying Bayesian inference (BI) with Mr. Bayes v.3.2 software [32]. Ten Markov chains were proceeded for ten million generations, and trees were sampled for every seven thousand generations, setting the program to run the substitution model with the option invgamma and permitting different rates of transition and transversion (K80 or HKY85 model). Twenty-five percent of the sampled trees were discarded, and the remaining trees were used to build up a consensus tree and to calculate the posterior probabilities of the clades. The results of the Bayesian analyses were visualized using Figtree v1.4.2. A transformation of the leaves and a schematic representation of the root were applied for visualization purposes.

### 2.8. Clonal Complex Determination

To infer patterns of evolutive descendants for the *L. panamensis* DSTs, we used the goeBURST algorithm, which is included in the bioinformatic package PHYLOVIZ v.2.0 [33]. The goeBURST algorithm predicts the offprint of a founder genotype, establishing groups of related strains that share a certain number of identical alleles with other members (clonal complex).

### 2.9. Global Ancestry of Local Leishmania panamensis Isolates

We arranged the haplotype information from each locus in a matrix in a single text file to estimate the ancestral population proportion for each *L. panamensis* isolate, using the software STRUCTURE v2.3.4 [34]. STRUCTURE grouped individuals with similar patterns of variation into populations of genetic groups and estimated global ancestry by applying different models of population structure to the data [35]. To estimate ancestry, we first estimated the most likely value of k (genetics groups) by running the software twenty times for values of k between 1 and 15. The software parameters were set to run the simulation for a burning period of 20,000 iterations and 60,000 Markov chain Monte Carlo repetitions using the admixture model of ancestry. The mean value and standard deviation calculated from the natural logarithm of the prior probability from each k repeat were used to assess the ∆K value (the rate of variation of the log-likelihood of data). This parameter was then applied to estimate the k value that best fits our data. We also employed the median value of Q (estimated membership) from each cluster of the most likely k obtained (k = 3) to summarize in a plot the estimation of this parameter for each local *L. panamensis* isolate. The percentage of membership for each isolate was represented in a stacked plot constructed in LibreOffice Calc and edited in LibreOffice Draw (https://www.libreoffice.org, accessed on 1 April 2022).

## 3. Results

### 3.1. Identification of Leishmania panamensis Isolates

*Leishmania* species were identified via Bayesian phylogenetic inference using a set of HSP70 sequences that are publicly available from the GenBank and TritrypDB databases (Appendix A). The isolates clustered into a clade conformed by the *L. panamensis* reference sequences used in the phylogenetic approach. Therefore, this result indicates that all local *Leishmania* isolates belong to this *Leishmania* species (Appendix A).

### 3.2. Genetic Diversity of L. panamensis Haplotypes

We successfully amplified the molecular markers included in the MLST approach in all local isolates of *L. panamensis*. Table 2 shows the diversity indexes from each molecular marker analyzed after constructing all *L. panamensis* haplotypes. The diversity indexes obtained suggest the presence of intraspecific variation in *L. panamensis* in Panama. The locus *GPI* and aconitase showed the highest diversity indexes, number of polymorphic sites, and haplotype diversity in this study. Consequently, we were able to determine six haplotypes via *GPI* and five haplotypes using aconitase. Conversely, the *HSP70* and *ALAT* loci showed the lowest theta values, Pi values, and haplotype diversity. However, we were able to distinguish three *L. panamensis* haplotypes with these loci. 

### 3.3. Diversity Indexes by Locus and Geographical Region

Panama presents ecological differences between the western and eastern sites of the country, particularly in the distribution of seasonal precipitation, the annual average of precipitation and daily fluctuations in temperature, and vegetal coverage that might influence the local transmission of leishmaniasis and consequently the populational structures of the parasites circulating in both regions. Therefore, we evaluated diversity indexes in the western and eastern regions of the country. After this evaluation, two loci indicated clear differences between the genetic diversity indexes found in the two regions (Figure 2, Table 3). The differences were more evident when we applied aconitase and *GPI* approaches to assess the *L. panamensis* haplotype diversity. Using these two markers, we found a higher diversity index in the *L. panamensis* population from the western region. With the *ALAT* approach, slight differences between the diversity indexes from the two areas were observed, while with the *HSP70* analysis, we detected no diversity in the east region, as only one haplotype was found circulating in this area. The number of polymorphic sites and haplotypes also supports the diversity differences between the local *L. panamensis* populations from both regions (Table 3). All diversity indexes evaluated in this study were demonstrated to be higher when *GPI* and aconitase were used as markers, and the highest values were demonstrated in the western region, where the *L. panamensis* populations appear to be more diverse.

Conversely, the *ALAT* approach found the same number of polymorphic sites and haplotypes in both regions, with two haplotypes defined by one single nucleotide polymorphism (SNP). On the other hand, the *HSP70*-selected fragment demonstrated the presence of only one haplotype in the east region and three haplotypes circulating in the west region due to the presence of two more SNPs in the *HSP70* locus of this population.

### 3.4. Genotyping Approach by MLST

To optimize the MLST approach, we first determined the best number and combination of loci that would produce the highest diversity values. We found that as the number of loci used in the MLST scheme was increased, there were increments in the minimum, maximum, and mean numbers of DSTs found by the algorithm of the MLSTest software. As a result, we decided to use the MLST scheme containing all four loci to find the highest intraspecific diversity in the *L. panamensis* isolates evaluated in this study. As a part of the optimization process, we also performed a congruency analysis for the four gene selections. After evaluating incongruency length differences and comparing the topology of the concatenated locus with the loci selected for the MLST approach, we found no incongruency as the *p*-value for the ILDP parameter and Templeton test was 1. Moreover, our MLST approach was able to determine thirteen *L. panamensis* DSTs because of a collective typing efficiency of 0.85, a combined discriminatory power of 0.491, and the presence of 20 SNPs considering all loci (Appendix A).

### 3.5. Geographic Distribution of DSTs

The analysis of the allele combination from the selected loci obtained in this study evidenced the circulation of thirteen *L. panamensis* genotypes that were named in a range from DST1 to DST13. Figure 1 depicts the geographical distribution of the thirteen DSTs found in the studied areas. The *L. panamensis* genotype DST1 was the most prevalent in all studied provinces from the western and eastern regions of the country. The second most frequent genotype was DST6, which was found in the provinces of Panama, Panama Oeste, and Cocle. The rest of the DSTs seem to have a restricted geographical distribution as they were found only in particular provinces. However, because a low number of isolates were evaluated in some provinces, we cannot rule out the presence of these DSTs or additional ones in other regions of the country. Genotypes DST2 and DST4 seem to be restricted to the northern region of Panama Province. Additionally, in the eastern region of Panama Province, we found DST6, DST7, DST8, DST9, and DST13, and nearby, in Darien Province, we found the genotype DST3 circulating. On the other hand, in the western region of the country, we found the genotypes DST7 and DST6 circulating in Panama Oeste, DST6, DST11, and DST10 circulating in Cocle Province, and genotypes DST5 and DST12 circulating in Bocas del Toro Province (Figure 1).

### 3.6. Haplotype Resolution of Local L. panamensis Isolates

To establish the haplotype compositions of the local *L. panamensis* isolates, we resolved the ambiguities found in all loci using the PHASE algorithm of the DNAsp software v6.12.03 (Appendix A). The isolates grouped into DST1 were shown to be homologous genotypes comprised of the same haplotype in all studied loci. The remaining genotypes were found to be heterologous isolates bearing one to three ambiguities in the one locus to three loci evaluated. The genotypes defined by the ambiguities located in the aconitase loci were DST5, DST6, DST7, DST9, and DST13. Only one ambiguity located in the *ALAT* locus discriminated DST3 from the rest of the *L. panamensis* genotypes. Regarding the *GPI* locus, the SNPs found in this locus made it easy to distinguish DST2, DST4, and DST8 from the other *L. panamensis* genotypes. The isolate FID16203-674, which belongs to DST12, was the only genotype defined by an ambiguity located in the *HSP70* locus. On the other hand, the heterologous isolates FID16203-453 (DST10) and FDI16203-454 (DST11) presented ambiguities in two and three loci, respectively. The isolate FID16203-453 showed ambiguities in the aconitase and *ALAT* loci, and the isolate FID16203-454 showed ambiguities in the aconitase, *GPI*, and *HSP70* loci.

The evidence found in this study suggests that most of the DSTs found herein are heterologous genotypes circulating at low frequencies in most of the endemic areas evaluated. Additionally, the clonal complex analysis indicates that these heterologous genotypes arose from a homologous genotype (DST1) that diversified via genetic exchange; nowadays, its present variant might participate in different transmission cycles. The results also suggest that in local areas such as Cocle Province (Appendix A) exists an intense genetic exchange between *L. panamensis* genotypes that are giving rise to new variants of this *Leishmania* species. This could be the case for the heterologous genotypes bearing ambiguities in two or more loci found in this study.

### 3.7. Phylogenetic Analysis and Clonal Complex Determination of L. panamensis DSTs

The phylogenetic analysis via Bayesian inference, using the concatenated sequences from the loci analyzed in this study, supported the existence of 13 *L. panamensis* DSTs that are depicted branching with high credibility values in Figure 3. The alleles from the *L. panamensis* DSTs were grouped together in a major clade with a credibility value of 0.98, corroborating that they are phylogenetically related, and the DSTs obtained represent different genotypes of *L. panamensis*. In addition, the phylogenetic analysis revealed a close association among the *L. panamensis* genotypes DST6, DST7, DST10, and DST11, which cluster together in the same clade. The clonal complex analysis supports this fact as it indicates that the genotype DST11 arose from DST6 and the genotype DST10 arose from DST7 (Figure 3). These genotypes overlap in distribution, and some of them seem to be circumscribed to endemic areas from Cocle Province, indicating this geographical point as a possible hotspot in which inbreeding events and genetic exchange are contributing to the high genetical variability found in the *L. panamensis* population from this area. 

Additionally, the clonal complex test performed on the sequences of the representative alleles from the *L. panamensis* DSTs suggests the presence of one *L. panamensis* clonal complex conformed by DST1 as a founder genotype and another 12 *L. panamensis* DSTs that differ in one or two loci from DST1 (Figure 3). 

### 3.8. Global Ancestry of Local L. panamensis Isolates

To better characterize the population structure of *L. panamensis* in the studied areas, we assessed the median value of the ancestral population proportion for each local isolate (Figure 4). Global ancestry estimates the proportion of ancestry from each contributing population, and each proportion can be considered an average over the individual´s entire genome [34]. Taking this fact into consideration, we estimated the contribution of each ancestral population to the genetic constitution of the local *L. panamensis* isolates. Interestingly, we found that the genetic make-up of all isolates, regardless of their geographic origin, is a product of the combination of different proportions of genetic segments from three ancestral populations. Furthermore, the local *L. panamensis* genotypes inferred by the program MLSTest showed unique patterns of the ancestral contribution that mirror the specific contribution of the three ancestral populations to each *L. panamensis* genotype studied herein. This fact also points out that in the case of admixture DSTs, genotype-specific patterns of the ancestral contribution could be used to identify different genotypes in endemic areas.

## 4. Discussion

*Leishmania* species are comprised of different populations bearing medically important characteristics that make their identification relevant in all epidemiological landscapes. Indeed, the genetic variability at the species level greatly impacts the biological diversity of its populations, influencing important properties such as virulency and tolerance to certain drugs. In this sense, it is highly recommended to type *Leishmania* parasites beyond the species level to properly understand their population structure, population dynamics, geographical distribution, and transmission cycles of medically relevant genotypes.

In this study, we used an MLST approach based on four housekeeping gene segments to infer the genetic diversity of *L. panamensis* at the haplotype and genotype levels and to determine the distribution of genetic variants of this species in Panama. We based the molecular characterization of the *L. panamensis* isolates on an MLST approach as it will probably become the recommended gold standard for *Leishmania* genotyping, including phylogenetic and epidemiological studies in different geographical areas [21]. In fact, this tool has been already used with *Leishmania* strain/isolates of the subgenus *Viannia* to study phylogeny and population genetics [20], discriminate *Viannia* species [36], investigate leishmaniasis outbreaks [25], and study parasite variability and the phylodynamics of this disease [20,25].

The present study represents a pioneering initiative aiming to assess the local genetic variability of *L. panamensis* in the country via MLST. For this purpose, we first evaluated the genetic variability of this species at the haplotype level. The combination of haplotype reconstruction by locus and the diversity index evaluation allowed us to assess local *L. panamensis* variation. All evaluated markers presented different degrees of polymorphisms mirrored by the differences in the diversity indexes and the number of haplotypes reported by locus in this study. This finding provided us with a preliminary glance at the local variation of *L. panamensis* and suggested a certain degree of intra-specific variability in this *Leishmania* species at the haplotype level. In fact, our results suggest differences between the haplotype diversity found in the local *L. panamensis* isolates from the eastern and western regions of the country. The western region showed higher haplotype diversity indexes and outnumbered the eastern region in the number of haplotypes. Differences in genetic diversity at the haplotype level between these two regions have also been reported for *Plasmodium vivax* [37]. In the case of *P. vivax*, it was suggested that different ecological and climate conditions as well as vector diversity prevalent in both areas have influenced local malaria transmission and therefore the genetic structure of *Plasmodium*. Indeed, the western and eastern regions have different ecological conditions, types of land covering, and weather conditions, such as seasonal distribution of rainfall, mean annual temperatures, and daily temperature fluctuations [38]. These differences in ecoclimatic conditions between areas might be shaping the genetic make-up of *L. panamensis* in Panama, as macroecological and climate conditions have been shown to influence the patterns of transmission and the seasonality of this disease in the country [23,39]. A study carried out in Colombia using an MLST approach also found geographical differences in the haplotype diversity of *L. panamensis* in regions with different ecoclimatic profiles [26]. Changes in climatic conditions and ambient temperature can affect the distribution of leishmaniasis through sand fly abundance or through the effect of temperature on parasite development in vectors [40]. In this regard, the capability of *L. braziliensis* to produce heavy, late infections in sand flies in the range of temperatures between 20 and 26 °C has been demonstrated [41]. This plasticity of the *Viannia* species might be one of the factors that drive the dissemination of parasites to different ecoclimatic areas and further divergency to produce different *L. panamensis* haplotypes in the Americas.

After combining the haplotype reconstruction algorithm of DNAsp and the capacity of the MLSTest software v1.0.1.23 to determine DSTs, we found more diversity in the local populations of *L. panamensis* by inferring the haplotype resolution at the diploid state of the parasite. Applying this approach, we found thirteen *L. panamensis* genotypes that cause TL in most of the endemic areas of the country. This finding, and the one obtained via haplotype analysis, confirms the circulation of genetic variants of *L. panamensis* in Panama. The phylogenetic analysis supports this result by grouping all DST alleles with *L. panamensis* reference sequences and confirming them as specific genotypes of this species that come from the same population. Few studies in Panama have described the genetic diversity of *L. panamensis*. A preliminary study using a microsatellite panel carried out in a small geographic region in Central Panama, revealed extensive genetic diversity in 27 *L. panamensis* isolates [22]. However, no repeated genotypes were detected in that study, revealing the high resolution of this set of microsatellites in such a small group of isolates. This fact might hinder a further association between local *L. panamensis* variants and the clinical expression of TL in the country, validating the probable need for less resolutive tools such as multilocus typing to address this question. A recent study that evaluated a set of SNPs from 24 *L. panamensis* genomes in Panamá found a genetically divergent group of isolates circulating in some endemic areas of the country, thus corroborating the presence of genetic variants of this parasite in Panama [42]. Recently, our research group reported three haplotypes of *L. panamensis* circulating in the country [12]. In that study, the typing approach was based on one molecular marker, a fact that might explain the lack of resolution in finding more *L. panamensis* haplotypes using a one-marker strategy. As previously mentioned, when evaluating levels of characterization in the sub-species category, it is recommended to add more loci to typing schemes in order to increase the discriminatory power [21]. In this regard, our MLST scheme, which was based on four housekeeping genes, supports the previous evidence regarding the circulation of *L. panamensis* variants in Panama and shows great resolution, identifying several genotypes of this species. Additionally, each local *L. panamensis* genotype presented a specific profile in the summary plot of estimates of Q, indicating the inheritance of different allelic proportions by the local *L. panamensis* genotypes. As these proportions are obtained as an average over the organisms’ genomes, this finding also supports the circulation of *L. panamensis* genotypes with different genetic make-ups in the local epidemiological scenario. These genetic differences might be further mirrored in different medically important characteristics such as the clinical expression of the disease, parasite internalization into macrophages, immune evasion, virulence, pathogenicity, and drug resistance. In this sense, the medical staff from the clinical branch of the Instituto Conmemorativo Gorgas de Estudios de la Salud, located in Panama City, has reported in its clinical practice human cases of localized, disseminated, diffuse, and mucocutaneous forms of TL associated with *L. panamensis* [15]. Such variability in clinical forms in its broadest sense could be the result of the local interplay between variants of this species and inhabitants with different genetic profiles. Therefore, it is important to carry out further studies using high-resolution techniques, such as the one developed herein, to unveil the association between *L. panamensis* make-up and the clinical expression of leishmaniasis in the country. In this regard, we are gathering all the necessary clinical information to evaluate the association between these two variables, using the molecular tools developed herein.

Regarding the distribution of *L. panamensis* in the studied areas, no specific pattern was found. It seems that DST1 is widely spread in the country as it was found in all studied areas. Unlike this genotype, the rest of the *L. panamensis* DTSs were less frequently found, and most of them circumscribed to specific geographical points at which DST1 was also reported. This association between DST1 and other *L. panamensis* genotypes in specific points of the country might indicate that DST1 diversified to produce new genotypes allied to specific transmission foci. The clonal complex analysis supports this assumption, indicating DST1 as a founder genotype that differentiates from the rest of the DSTs by one to three allele differences. This result suggests that DST1 might represent the ancestral genotype from which the rest of *L. panamensis* DSTs emerged in the country. This genotype could have arrived in the country after the formation of the isthmus that prompted the settlement of the vectors and reservoirs of *Leishmania* parasites. In fact, Valderrama and collaborators [43] proposed that *Lutzomyia gomezi*, one of the most important local vectors, was established and disseminated in the country after the closure of the Isthmus. The same authors also stated that several mammals’ reservoirs of *Leishmania* parasites settled in the country after this event. After that, the spreading of DST1 to areas presenting different ecoclimatic conditions with particular vectors and reservoirs’ host compositions might have contributed to the emergence of new local *L. panamensis* genotypes. Furthermore, geographical barriers in Panama, such as the Central Mountain range, possibly restricted the *Lutzomyia* populations to specific demographic areas in which different *Leishmania* cycles were established. This circumstance might have driven the local *L. panamensis* diversification and expansion as it was suggested elsewhere for the *Lu gomezi* vector [43]. 

As for the low frequency and geographical constraint of most of the *L. panamensis* genotypes, this fact could be the result of geographical isolation, followed by divergence from DST1. Additionally, these genotypes might be participating in specific transmission cycles along with particular vectors and reservoirs circumscribed to specific transmission foci. *Leishmania* strains structure at a small geographic scale and patterns of structuration have been reported for *L. guyanensis* in French Guyana [44] and *L. braziliensis* in Peru and Bolivia [45]. In such a transmission focus, *Leishmania* vectors and reservoirs might play an important role in shaping the genetic make-up of the *Leishmania* species. Sexual events happening in vectors represent a great source of *Leishmania* diversity, and the clonal expansion and dissemination of the parasite populations are ensured by mammal hosts. In Panama, *Choloepus hoffmani* is the main reservoir host of *L. panamensis* [46,47], and its presence might be associated with high human population densities, facilitating the local transmission of TL [48]. Additionally, secondary mammal reservoirs responsible for the maintenance of the parasite population in enzootic cycles have also been described in the country [49,50]. Anthropophilic vectors participating in both enzootic and domestic/peridomestic cycles because of their adaptation to disrupted environments might be responsible for introducing new genotypes to local endemic areas [51]. This could be the case for the *L. panamensis* genotypes found in this study at low frequencies across the country.

Our study presented a few important limitations that need to be stated. First, the study sample size was rather small, and therefore, our results do not necessarily represent the entire genetic diversity of *L. panamensis* that could be seen in the country or in the endemic regions evaluated. For this reason, we cannot rule out the presence of other genotypes in regions where they were not identified, although the overall patterns of increased diversity in western Panama when compared with eastern Panama are likely robust. Second, we did not address the temporal distribution of DSTs; thus, it is unknown whether these genotypes are totally fixed in the local *L. panamensis* populations and have enough continuity throughout time to be responsible for the local cases of TL during all seasons. Indeed, the temporal presence of the *L. panamensis* DSTs found herein must be evaluated before considering them in any local programs for leishmaniasis control and surveillance.

The global ancestry result inferred by STRUCTURE showed that all *L. panamensis* genotypes found in this study are admixture individuals that originated from three ancestral populations. A recent analysis of medically relevant SNPs from a small sample of 24 *L. panamensis* isolates from Panama also found admixture individuals [42]. Events of endogamy are common in *Leishmania* parasites and recurrently happen in species from the same *L. panamensis* subgenus, such as *L. guyanensis* [44] and *L. braziliensis* [45]. Inbreeding also was suggested for *L. panamensis* during a study that analyzed a set of microsatellites in 27 isolates of this species from regions located in Central Panama [22].

Despite *L. panamensis* being an important etiological agent of TL in Mesoamerica and some South American countries [52], few studies have been conducted to explore the genetic diversity of this *Leishmania* species. A study carried out in Colombia via MLST obtained a greater diversity index using the *ALAT* locus to evaluate the genetic diversity of local populations of *L. panamensis* than the one inferred for us using the same marker. These variations in the diversity index could be explained by either a difference in sample size between studies or geographical isolation and the independent evolution of both *L. panamensis* populations. As a matter of fact, a recent study using 43 isolates of Colombian and Panamanian origins found that the Panamanian population of *L. panamensis* is genetically divergent from its Colombian counterpart [42]. As parasites from the *Viannia* subgenus structure in small geographical areas [44], it is not rare to find divergent groups of *L. panamensis* separated by distance and geographical barriers, such as the ones existing between Panama and Colombia. Furthermore, geographic isolation and the restricted genetic flow of parasites due to the low dispersion of infected vectors and mammals might be favoring the structuration of *Leishmania* populations at a small scale and the emergence of genetic variants constrained to specific endemic areas. This could be the case for the genetic variants of *L. panamensis* found herein and in other studies carried out in Panama [12,22,42] and Colombia [26] and probably in other Latin American countries where this species circulates. This phenomenon might be potentiated by the presence of highlands and mountain ranges along the endemic territories that have facilitated isolation, genetic divergence, and speciation events. For instance, the Mesoamerican region is considered to be the second most-threatened biodiversity hotspot in the Americas [53]; it is highly fragmented, like an archipelago, and its taxa are thus frequently represented as sets of isolated populations, each restricted to particular mountain ranges and often showing a high degree of divergence, both morphologically and genetically [54].

Even though *L. panamensis* has shown a lower level of genomic variability than members of the *Viannia* subgenus, such as *L. braziliensis* [55], the polymorphisms found in a selected group of loci via MLST and the evaluation of selected sets of SNPs at the genomic level in Latin America [26,42] indicate that intra-specific genetic variability exists in *L. panamensis*. Other members of the *L. guyanensis* complex, such as *L. guyanensis* and *L. naiffi*, have shown low levels of structural variations and high degrees of structural homogeneity among *Leishmania* strains of the same species regardless of the heterogeneity observed in the selected sets of SNPs/Indels analyzed [56]. It has been hypothesized that the heterogeneity of such SNPs might be attributed to the parasite’s adaptation to new human hosts and different ecological niches [56]. Consequently, the diversity in human population groups and ecological niches present in Latin American countries where *L. panamensis* is circulating might be shaping the genetic structure of this parasite, somehow causing the emergence of genotypes restricted to the epidemiological scenarios characteristic of each country. The structuration of *L. panamensis* into subgroups, variants, or lineages at a small geographical scale highlights the necessity of developing regional initiatives aimed at revealing how the genetic composition of this species impacts medically relevant characteristics of the parasite that might be driving the disease outcome in Latin America.

## 5. Conclusions 

As far as we are concerned, this is the first study describing the genetic diversity of *L. panamensis* in most of the endemic regions of the country. Our results clearly indicate the circulation of thirteen *L. panamensis* genotypes belonging to the same population associated with TL. Moreover, these results support the high level of intra-specific genetic diversity found in *L. panamensis* using SNPs as a marker using either MLST or NGS approaches in other Latin American countries, such as Colombia. Our findings pave the way for developing further studies aimed at shedding light on the relationship between the genetic composition of *L. panamensis* and the important, medically relevant characteristics of this parasite. Furthermore, the MLST approach developed in this study could be used as a feasible option for typing and inferring the genetic structure of *L. panamensis* in developing countries of Latin America, where obtaining complete genomes of this species involves higher costs.

## Figures and Tables

**Figure 1 pathogens-12-00747-f001:**
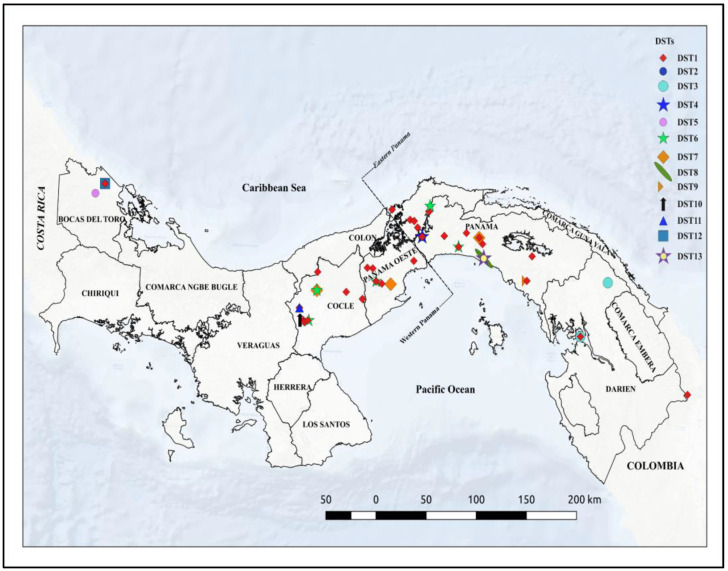
Map of Panama showing the geographical distribution of the local *Leishmania panamensis* diploid sequence types (DSTs) found in this study.

**Figure 2 pathogens-12-00747-f002:**
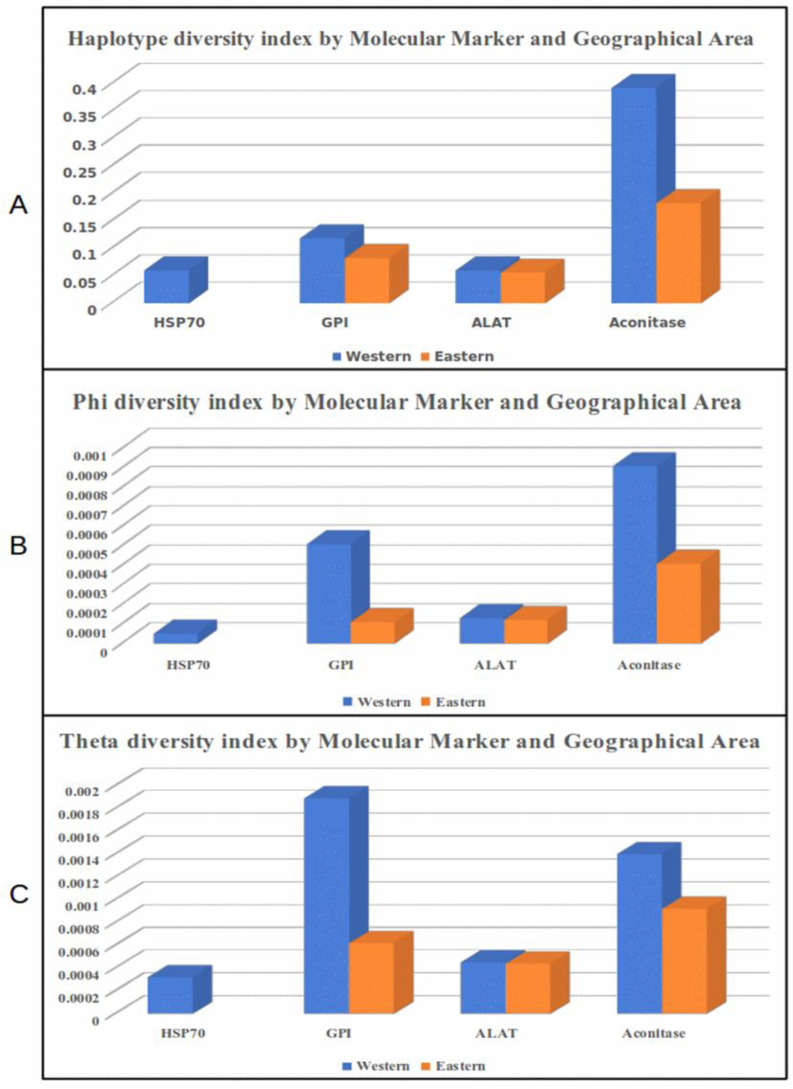
*Leishmania panamensis* diversity index by molecular marker and geographical area in Panama evaluated in this study. (**A**) Haplotype diversity index; (**B**) phi diversity index; (**C**) theta diversity index.

**Figure 3 pathogens-12-00747-f003:**
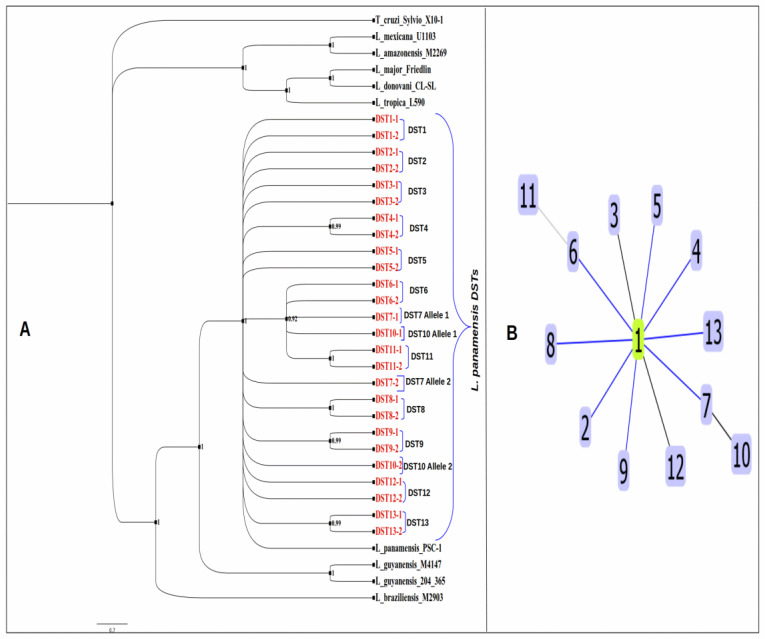
(**A**) Bayesian phylogenetic tree inferred using the concatenated sequences of each DST allele from the *L. panamensis* found in this study. The code of each DST allele is highlighted in red. (**B**) Clonal complex analysis showing the founder (DST1) in green and its genetically related groups, DST2–DST13, which are represented by the numbers from 1 to 13.

**Figure 4 pathogens-12-00747-f004:**
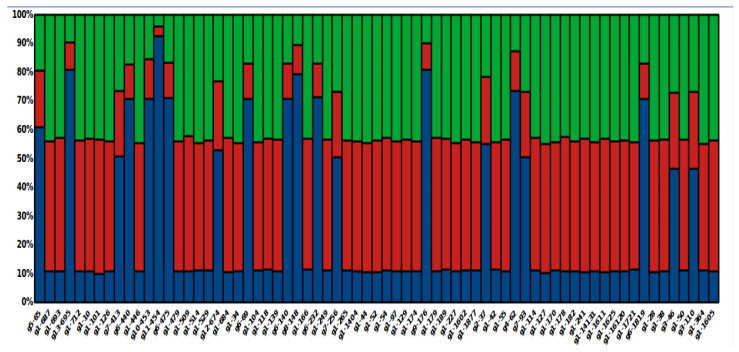
Global ancestry analyses of each *L. panamensis* isolate obtained in this study. The name observed on the *X*-axis is a composed code that stands for the *L. panamensis* genotypes (letters G1–G13) and the *L. panamensis* isolation code (numbers after the dashes).

**Table 1 pathogens-12-00747-t001:** List of gene loci and primers used in this study.

Target Gene	Enzyme Entry	Gene Length (bp)	Chromosomal Location	Amplicon Size (bp)	Primers Sequences	Reference
Heat Shock Protein 70	AAG01344.1	2566	28	1364	PLeishF: GATGGTGCTGCTGAAGATGA PLeishR: GGTCATGATCGGGTTGCATR	This study
Glucose-6-phosphate isomerase	EC 5.3.1.9	2084	12	1745	GPIextF: AAT GTT CTT CAT ACC CCT CT GPIextR: TTC CGT CCG TCT CCT G GPIintF: TGG GAT TGG CGG CAG CGA CCTT GPIintR: CGC CAC AGG TAC TGG TCG T	[27]
Alanine aminotransferase	EC 2.6.1.21	1493	12	589	ALAT.F: GTGTGCATCAACCCMGGGAA ALAT.R: CGTTCAGCTCCTCGTTCCGC	[20]
Aconitase	EC 4.2.1.3	2690	18	579	ACO.F: CAAGTTCCTGRCGTCTCTGC ACO.R: GAGTCCGGGTATAGCAKCCC	[20]

**Table 2 pathogens-12-00747-t002:** Genetic diversity of local *Leishmania panamensis* population by locus evaluated.

Loci	s	h	hd	π	θ
*HSP70*	2	3	0.029	0.00002	0.00028
*GPI*	12	6	0.099	0.00030	0.00217
*ALAT*	2	3	0.057	0.00012	0.00077
Aconitase	4	5	0.290	0.00067	0.00162

s: segregations sites; h: number of haplotypes; hd: haplotype diversity; π: pi: diversity index; θ: theta diversity index.

**Table 3 pathogens-12-00747-t003:** Genetic diversity of local *Leishmania panamensis* populations by molecular marker and geographical area.

	Western N = 34					Eastern N = 35				
Marker	s	h	hd	π	ϴ	s	h	hd	π	ϴ
*HSP70*	2	3	0.060	0.00005	0.00032	0	1	0	0	0
*GPI*	9	4	0.118	0.00051	0.00189	3	3	0.082	0.00011	0.00062
*ALAT*	1	2	0.060	0.00013	0.00045	1	2	0.055	0.00012	0.00044
Aconitase	3	4	0.392	0.00091	0.00140	2	3	0.182	0.00041	0.00092

s: segregations sites; h: number of haplotypes; hd: haplotype diversity; π: pi diversity index; θ: theta diversity index, N = number of *L. panamensis* isolates.

## Data Availability

Individual alignments of each locus used in this study as well as the concatenated alignments containing sequences from aconitase, *ALAT*, *GPI*, and *HSP70* loci are available upon request. The sequences obtained in this study are publicly available in GenBank database and can be retrieved using the accession numbers described in the Materials and Methods section of this study.

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
