# Peer review of "Insights into the Genetic Diversity of Leishmania (Viannia) panamensis in Panama, Inferred via Multilocus Sequence Typing (MLST)"

_pathogens, 2023, doi:10.3390/pathogens12050747_

Round 1
Reviewer 1 Report
The manuscript is written well in general and I understand the importance of genotyping local species of Leishmania, but the strength of the presented study is not very clear in the current version. The authors are encouraged to give more information on the obtained data and restructure the manuscript.
Major comments
Can you provide more information on the selected genes and the reasons why they were selected? Required information includes, but not limited to, chromosome number for each gene. These should be written not in Discussion section but in Introduction or Materials & Methods section.
People often use ITS but not protein-coding sequences for high resolution genotyping due to relatively higher diversity in non-coding ITS. Similar rationale may apply for the choice of microsatellite as reported previously and cited in this manuscript. In fact, Restrepo et al. used 17 microsatellite loci for genotyping. Can the authors provide scientific evidence that their approach provides better resolution than the previous work performed in Panama?
Table 2: It is not clear how many isolates are from Western and Eastern Provinces respectively, therefore making the interpretation of the data difficult.
Table 2: It is also difficult to understand the numbers in the table. The table says the numbers of haplotypes identified by ALAT are 2 in Western and 2 in Eastern. However, the text says 6 haplotypes are identified by the gene (L254). The same applies to the other genes. In addition, Table S2 says there are 3 alleles for ALAT, but I don’t see information if strains examined were heterologous or homologous. Can you explain, if my interpretation is wrong, more about this so that the readers understand the data more easily?
Table 2: Is there any statistical analysis on the difference in diversity between Western and Eastern Provinces?
In conclusion, what is the benefit of the study? In other words, by identifying the presence of genetic diversity in L. panamensis in Panama, how can you connect the knowledge to understanding of other things? For example, did the authors find any connections between genotypes and virulence? Is genetic diversity in Panama larger or smaller than other countries, or is L. panamensis more stable/unstable compared with other species?
Related to the above comment, the Discussion section is lengthy but lacks deep discussion on the importance of the findings. Thorough restructure of the Discussion section is strongly encouraged.
Minor comments
L3: Small letter for P of panamensis
L69: What do you mean with ‘gender’?
L104: What does DST stand for?
Author Response
point 1
Can you provide more information on the selected genes and the reasons why they were selected? required information includes, but is not limited to, the chromosome number for each gene. This should be written not in the Discussion section but in the Introductions of Materials & Methods sections.
The authors agree with the editor’s comment therefore we added all the information regarding the gene selection in the methodology section of the manuscript (lines 149 to 163). Also, we added information on chromosomal location, gene length, and product size to Table 1.
Point2
People often use ITS but not protein-coding sequences for high-resolution genotyping due to relatively higher diversity in non-coding ITS. A similar rationale may apply to the choice of microsatellite as reported previously and cited in this manuscript. In fact, Restrepo et al used 17 microsatellites loci for genotyping. Can the authors provide scientific evidence that their approach provides better resolution than the previous work performed in Panama?
We agree that ITS and microsatellites have a higher rate of evolution than protein-coding regions and thus provide a higher resolution for genotyping. However, the authors believe as stated in lines 483-485 that approaches based on less resolutive markers as the ones used in our study might correlate better medically relevant characteristics of L. panamensis with its genetic composition. Restrepo et al analyzed 27 isolates from Central Panama and all of them were found to be new L. panamensis variants due to the high resolution of their microsatellites approach. We think that this finding might hinder the correlations between variants and the clinical expression of the disease, for example.
Point 3
It is not clear how many isolates are from Western and Eastern Provinces respectively, therefore making the interpretation of the data difficult.
The authors thank the reviewer for the comment. To clarify the number of isolates coming from both Panamanian regions, we added to Table 3 the number of L. panamensis isolates belonging to each region.
Point 4
It is also difficult to understand the numbers in the table. The table says the number of haplotypes identified by ALAT is 2 in Western and 2 in Eastern. However, the text says 6 haplotypes are identified by the gene. The same applies to the other genes. In addition, Table S2 says there are 3 alleles for ALAT, but I don’t see information if strains examined were heterologous or homologous. Can you explain, if my interpretation is wrong, more about this so that the readers understand the data more easily?
We agree with the reviewer’s comment so to clarify the data shown in Table 2, we split the data into two tables. Table 2 summarized the haplotype diversity found in all studied areas and Table 3 described the diversity indexes found by geographic area. Lines 194 to 197 of the methodology section described how we assessed diversity indexes from 34 and 35 L. panamensis isolates belonging to the western and eastern regions. Regarding the homologous and heterologous states of the haplotypes, we added the supplementary table S3 to the manuscript describing the haplotype resolution of each isolate found in our study mentioning the results of this analysis in subsection 3.6 of the results (lines 348 to 363).
Point 5
Table 2: Is there any statistical analysis on the differences in diversity between Western and Eastern Provinces?
We thank the reviewer for the comment. We believe that one of the limitations of our study is the sample size (34 and 35 isolates from Western and Eastern regions, respectively). Due to this fact, we performed no statistical analyses comparing the diversity indexes from both regions.
Point 6
In conclusion, what is the benefit of the study? In other words, by identifying the presence of genetic diversity in L. panamensis in Panama, how can you connect the knowledge to the understanding of other things? For example, did the authors find any connections between genotype and virulence? Is genetic diversity in Panama larger or smaller than other countries, or is L. panamensis more stable/unstable compared with other species?
We agree with the reviewer’s comments and therefore added a conclusion section to highlight the benefit of this study (lines 613-624). Also, we discuss important aspects on the circulation of genetic divergent populations of this species and variants in the discussion section of the manuscript (lines 568-610).
Point 7
Related to the above comment, the discussion section is lengthy but lacks a deep discussion on the importance of the findings. Thorough restructuring of the discussion section is strongly encouraged.
We thank the reviewer’s comments and thus decided to review and restructure some points of the Discussion that was not totally addressed in this section.
All minor spelling corrections were addressed by the authors.

Reviewer 2 Report
This study deals with the analysis of 69 Leishmania panamensis strains by multilocus sequence typing (MLST) using 4 loci. The methodology is classical and sound. The study has mainly a local relevance, since this species' genetic variability is poorly known. The paper deserves to be published. I have a few remarks:
(1) what is "DST"? It is defined nowhere. I guess it has something to do with "sequence type" and would design MLST multilocus genotypes.
(2) I would disagree with the observation that adding new loci "do not necessarily increase the power of resolution of MLST". The use for bacteria is at least 7 loci. Some studies include much many more loci, with each time a muc higher resolution.
(3) The authors do not mention the adding of outgroup species in their study, although they certainly did it as shown on supplementary figure 1.
Author Response
Point 1
What is DSTs? It is defined nowhere I guess it has something to do with sequence type and would design MLST multilocus genotypes.
We agree with the reviewer’s comment so to clarify the meaning of DSTs, we added to the methodology section of the manuscript (lines 103 to 104), what this abbreviation stands for.
Point 2
I would disagree with the observation that adding new loci “do not necessarily increase the power of resolution of MLST. The use for bacteria is at least 7 loci. Some studies include much many more loci, with each time a much higher resolution.
We agree with the reviewer’s comment and therefore we omitted the line stating that increasing the number of loci beyond four genes does not necessarily increase the discriminatory power of MLST from the Discussion section of the manuscript.
Point 3
The authors do not mention the adding of outgroup species in their study, although they certainly did it as shown in supplementary figure 1.
We agree with the reviewer´s comment, thus, we decided to mention that we used an outgroup in the methodology section of the manuscript (lines 141 to 143).
Round 2
Reviewer 1 Report
The authors have responded to the reviewer’s comments.
Point 5: The authors’ answer to my comment is about the limitation of their study, i.e., the small sample size. I understand the situation but it cannot be a reason why they skipped statistical analyses. At least, the authors need to describe the limitation to the reader if they are aware of it.
L158 and L597: intra-species?
Author Response
Response to the Reviewer
Comment 1
The authors’ answer to my comment is about the limitation of their study, i.e., the small sample size. I understand the situation but it cannot be a reason why they skipped statistical analyses. At least, the authors need to describe the limitation to the reader if they are aware of it.
We agree with the reviewer's comment and to highlight the limitations of our study we added a paragraph in lines 557-570 talking about these limitations.
Comment 2
L158 and L597: intra-species?
To clarify the reviewer's comment, we re-wrote lines 158-157 in this way:
"Additionally, aconitase, ALAT, and GPI loci have shown several polymorphisms that aided to explore the intraspecific or interspecific genetic variability of Leishmania species at haplotype and genotype levels when using reference sequences and/or sequences of Leishmania isolates from some Latin American countries".